# Alterations in Body Composition Lead to Changes in Postoperative Outcome and Oncologic Survival in Patients with Non-Metastatic Colon Cancer

**DOI:** 10.3390/jcm14103438

**Published:** 2025-05-14

**Authors:** Markus Philipp Weigl, Benedikt Feurstein, Patrick Clemens, Christian Attenberger, Tarkan Jäger, Klaus Emmanuel, Ingmar Königsrainer, Peter Tschann

**Affiliations:** 1Department of General and Thoracic Surgery, Academic Teaching Hospital, 6800 Feldkirch, Austria; 2Department of Radio-Oncology, Academic Teaching Hospital, 6800 Feldkirch, Austria; 3Institute of Medical Physics, Academic Teaching Hospital, 6800 Feldkirch, Austria; 4Faculty of Medical Sciences, Private University in the Principality of Liechtenstein (UFL), 9495 Triesen, Liechtenstein; 5Department of Surgery, Paracelsus Medical University Salzburg, 5020 Salzburg, Austria

**Keywords:** nutrition, colon cancer, body composition

## Abstract

**Background/Objectives**: Opinions concerning the role of obesity and changes in muscle mass in individuals with malignancies vary. It is believed that decreased fat tissue leads to a higher complication rate, while decreased muscle mass results in a poorer oncologic outcome. This study aimed to evaluate the impact of fat distribution and skeletal muscle mass on postoperative morbidity and long-term oncological outcomes in patients with non-metastatic colon cancer. **Methods**: Between 2012 and 2018, a total of 129 patients with stage I-III colon cancer were evaluated. Abdominal CT scans were used to assess muscle mass and fat tissue (subcutaneous and visceral). Differences in postoperative morbidity and long-term oncologic outcome were analyzed and compared. **Results**: No significant differences occurred concerning complication rate or anastomotic leakage. Individuals with altered body composition parameters had a shorter length of hospital stay (*p* = 0.046) and an increased duration of surgery (*p* = 0.029). In patients with an ASA III score, altered fat tissue distribution was associated with improvements in both overall and disease-free survival (*p* = 0.031 and *p* = 0.027, respectively) but also resulted in longer hospital stay. **Conclusions**: Changes in body composition parameters lead to alterations in economic factors as well as changes in oncologic survival, especially in patients with higher ASA scores. No differences in morbidity were observed.

## 1. Introduction

Colorectal cancer (CRC) is the third most common malignancy worldwide, accounting for over 10% of all diagnosed cancers [1]. Colon cancer constitutes approximately 60% of newly diagnosed CRC cases [2].

In non-metastatic carcinomas of the colon, the recommended treatment involves surgical removal of the affected bowel segment, including complete mesocolic excision, simultaneously removing local lymph nodes. Contrary to patients with rectal cancer, neoadjuvant (radio)chemotherapy is not the standard treatment for individuals with colon cancer. Depending on the nodal status, adjuvant chemotherapy may be indicated.

Anastomotic leakage (AL) is the most feared complication in colon surgery, with an incidence rate ranging from 2% and 19%. It is associated with increased mortality and morbidity, as well as prolonged hospital stays [3,4]. Common risk factors for AL include male sex, age, pre-existing comorbidities and high American Society of Anesthesiologists (ASA) fitness grade [5,6]. These are mainly non-influenceable variables. One aspect of patients’ characteristics that patients can influence themselves is body composition, especially the amount of fat tissue and muscle mass. While body mass index (BMI) is generally used to classify a person as underweight or overweight, it only considers height and weight to assess obesity, without accounting for the distribution of muscle and fat [7,8]. Numerous studies have shown that alterations in skeletal muscle mass and elevated levels of visceral and total fat are associated with increased postoperative complications and worse oncologic outcomes [9,10,11,12,13,14].

Nevertheless, no definitive opinion regarding the role of body composition and its influence on postoperative complications and survival in colon cancer patients could be made up to now.

The aim of this study was to investigate the influence of overall fat tissue, fat distribution and skeletal muscle mass on postoperative outcomes and oncological survival in patients with non-metastatic colon cancer.

## 2. Materials and Methods

Primary and secondary endpoint

The primary endpoint of this study was to assess the impact of body composition parameters (BCPs) on the rate of postoperative complications, especially anastomotic leakage. Secondary endpoints included the effects of these parameters on length of hospital stay, duration of surgery and oncologic outcome (overall and disease-free survival).

b.Inclusion and exclusion criteria

Patients with non-metastatic colon cancer (appendix, cecum, colon ascendens, colon transversum, colon descendens and sigmoid; UICC I-III) who underwent surgery between 1 January 2012 and 31 December 2018 were included in this study. A CT scan of the abdominal and pelvic region, conducted no more than 30 days prior surgery, had to be available.

Exclusion criteria consisted of the following: emergency surgery, metastatic disease, loss to follow-up and no available CT scan.

c.Preoperative evaluation

Prior to surgery, each individual received a CT scan of the trunk, as well as a colonoscopy with tissue biopsy. The therapeutic concept was discussed in a multidisciplinary team (MDT) and followed international guidelines.

d.Parameters

Demographic parameters included age, sex, height, weight, BMI, ASA classification [15], comorbidities—evaluated using the Charlson comorbidity score [16]—preoperative tumor staging, preoperative therapy, surgical technique, type of anastomosis, conversion (with reason for conversion), duration of surgery, complication (using the Clavien–Dindo Classification [17], length of hospital stay, disease-free survival and overall survival.

A subgroup analysis was performed on patients with an ASA score of III to assess the impact of BCPs on those with a more compromised health status.

Besides the height, weight and calculated BMI, further parameters were used to evaluate the body composition of the patients. They were as follows: total fat area (TFA, cm^2^), visceral fat area (VFA, cm^2^), subcutaneous fat area (SFA, cm^2^), skeletal muscle area (SMA, cm^2^) and skeletal muscle index (SMI; SMA/height2; cm^2^/m^2^). Furthermore, ratios between VFA and TFA as well as SFA and TFA were evaluated.

The listed parameters were classified using defined cut-offs (Table 1).

e.Histopathological examination

After surgery, the removed specimens were fixed in formalin. A macroscopic description was performed, followed by a complete histopathological examination using the TNM classification [20].

f.CT analysis

CT scans performed within 30 days prior to surgery were reviewed. All images were retrieved using the software Deep Unity Diagnost (DH Healthcare GmbH, Bonn, Germany Version 1.2.0.1). A single axial image at the level of the umbilicus was exported to a 3D imaging program (Horos TM, New York, NY, USA, Version 3.3.6) to evaluate the body composition parameters.

Different Hounsfield unit (HU) thresholds were applied to assess various fat and muscle distribution parameters. HU of −190 to −30 for TFA, VFA and SFA, HU of −30 to +110 for SMA and SMI.

To define sarcopenic patients, two criteria were used. The first margin was an SMA or SMI two standard deviations below the mean, the second being a gender-specific value (SMA/SMI ≤ 5th percentile) [21]. Sarcopenic obesity was defined as patients with an SMA below the cut-off and VFA above the respecting cut-off.

g.Sample size calculation

A sample size calculation was conducted using a confidence level of 95% and a margin of error of 0.05. With the available population in our region, the sample size calculation resulted in 384 individuals.

h.Statistical analysis

Statistical analyses were performed using Python (Python Software Foundation, Wilmington, DE, USA, Version 3.9.1). Continuous data were analyzed using the Mann–Whitney U-Test, Kruskal–Wallis Test or *t*-Test, as appropriate. Normally distributed data were expressed as mean ± standard deviation. Categorical variables were analyzed using the Chi-square test and presented as absolute numbers and percentages.

Uni- and multivariate analyses were performed to evaluate the effects of body composition on the entire cohort and a subgroup of multimorbid patients (ASA III). The multivariate analysis was conducted using a cox-regression analysis. Patients with sarcopenic obesity were compared to individuals without a combination of low SMA and high VFA. Statistical significance was defined as a *p*-value < 0.05.

i.Follow-up

The follow-up on patients was conducted via visits in our outpatient clinic following the recommended schema of the local oncologic authorities. Disease-free survival and overall survival were evaluated starting from the day of surgery and continuing until disease recurrence or death occurred, respectively.

## 3. Results

A total of 129 individuals were evaluated. The majority (77; 59.69%) were male, with a mean of 71.21 ± 11.94 years. The most common ASA classification score was III, while the average Charlson Comorbidity Index was 4.29 ± 1.62 points. A slight majority of all patients were classified as UICC Stage II (51; 39.53%). A total of 37 individuals (28.86%) received adjuvant chemotherapy.

More than half of the cohort underwent open surgery (70; 54.26%) with the most frequent surgical technique being the right hemicolectomy (61; 47.29%), followed by sigmoid resections (32; 24.81%) and left hemicolectomies (18; 13.95%). Of these 70 operations, 11 (8.53%) were converted from laparoscopic surgery. The mean duration of surgery was 146.37 ± 61.06 min.

Less than a quarter (30; 23.26%) of patients experienced postoperative complications, with bowel obstructions (7; 5.43%) being the most common. A total of four patients (3.10%) experienced anastomotic leakage. Among patients with complications, grade III according to the Clavien–Dindo Classification was the most frequent.

The average length of hospital stay was 13.3 ± 6.33 days.

Regarding body composition parameters (BCPs), the average height and weight were 1.71 ± 0.1 m resp. 77.2 ± 18.54 kg with a mean BMI of 26.29 ± 4.87 kg/m^2^. For further information regarding the remaining BCPs and general demographics, see Table 2.

The mean overall survival was 70.45 ± 31.19 months, while the disease-free survival was 67.48 ± 32.87 months. Additionally, the median overall survival was 76 months, the median disease-free survival 70 months.

Regarding the primary endpoint, no significant differences were observed in the overall cohort concerning complication rate or anastomotic leakage between individuals with normal and altered body composition parameters. For instance, the overall complication rate in patients with elevated TFA was comparable to those without (10; 27.8% resp. 20; 21.5%). Even more similar postoperative complication rates were found in individuals with a lower SMA (5; 23.8% resp. 25; 23.1%).

However, the length of hospital stay was significantly shorter in patients with elevated total fat area (11.92 ± 4.82 cm^2^ vs. 14.21 ± 7.08 cm^2^; *p* = 0.046). Furthermore, an elevated SFA/TFA ratio was associated with a longer duration of surgery (152.57 ± 63.20 vs. 123.67 ± 48.28; *p* = 0.029).

In the subgroup analysis of patients with an ASA III score, several composition parameters yielded significant results. Increased TFA was associated with improved overall (75.04 ± 28.88 months vs. 56.70 ± 33.77 months; *p* = 0.031) and disease-free survival (73.83 ± 30.48 months vs. 54.37 ± 34.54 months; *p* = 0.027). Elevated SFA was linked to a significantly longer duration of surgery in the whole cohort (open and laparoscopic) (159.58 ± 58.02 min vs. 119.41 ± 41.04 min; *p* = 0.002). Increased VFA was associated with a longer length of hospital stay (11.97 ± 4.48 days vs. 16.17 ± 8.79 days; *p* = 0.019) as well as an improved overall survival (74.06 ± 30.56 months vs. 53.37 ± 32.65 months; *p* = 0.01) and disease-free survival (71.19 ± 33.19 months vs. 52.26 ± 33.11 months; *p* = 0.024).

Regarding SMA and SMI, neither in the general cohort nor in the subgroup analysis could significant results be found.

Furthermore, the multivariate analysis and the analysis of patients with sarcopenic obesity did not result in any significant results (Table 2, Table 3, Table 4, Table 5 and Table 6 and Figure 1).

## 4. Discussion

Altered body composition is believed to affect postoperative morbidity and oncologic outcome in patients with colorectal cancer. In contrast to a previous study conducted by our study team on rectal cancer resection [22], body composition parameters have a different impact in patients with colon cancer. In this study cohort, elevated total fat appears to improve complication parameters, as well as oncological outcomes in subgroups. Another study examined the effects of changes in muscle mass in patients with rectal cancer undergoing radiochemotherapy (RCT), finding that a decrease in skeletal muscle during therapy was associated with the worst oncologic outcomes [23]. While RCT is commonly used in locally advanced rectal cancer, it is not the standard in colon malignancies, which may explain the differences in those two entities.

This study evaluated the effect of altered distribution of fat and muscle tissue in patients with non-metastatic colon cancer and its correlation to postoperative morbidity and oncologic outcome.

Two independent studies [24,25] suggested that increased visceral fat elevates the risk for anastomotic leakage. However, our findings did not confirm that hypothesis, as no significant differences were observed in patients with increased total fat, visceral fat or subcutaneous fat.

The impact of elevated TFA and/or BMI on length of hospital stay has been inconsistent in previous studies. Some studies show a decrease in hospital stay [26], while others show no or non-significant impacts. Our results indicate that an elevated TFA leads to a significant improvement in length of hospital stay. On the other hand, in the subgroup analysis of patients with an ASA III score, individuals with an elevated VFA spent significantly longer times in hospital. One reason could be the intraoperative challenges of a higher VFA, thus leading to more intraabdominal trauma and extended postoperative pain therapy. The finding was observed in the ASA III subgroup, which includes individuals with a compromised general health status and/or multiple comorbidities, potentially leading to a different response compared to the overall healthier study population.

Although no significant differences in oncologic survival were found in the total cohort, changes were observed in individuals with an ASA score of III. Increased TFA and VFA were associated with superior overall and disease-free survival, further supporting the inconsistency in the impact of elevated BMI or distribution of fat tissue on the recurrence of colon cancer [27,28].

Similar to findings regarding higher BMI, an elevated SFA/TFA ratio was associated with a longer duration of surgery [29], likely due to constraints in maneuverability and additional time required to open and close the abdominal wall. In addition, increased SFA in ASA III patients resulted in a longer operation time.

In contrast to a study conducted by Fleming et al. [18], no differences could be observed in patients with a high VFA/TFA or low SMA regarding the oncologic outcome, as no changes in rate of local recurrence, metastasis or overall/disease-free survival occurred. Therefore, their hypothesis concerning the role downregulation of anti-inflammatory and upregulation of pro-inflammatory cytokines could be not confirmed in this study.

Several limitations concerning our results do exist. First, because of the study’s retrospective nature, selection bias is a concern. During the selection process, multiple patients had to be excluded due to non-available CT scans. The total number of patients and the single-center nature could lead to the reduced power of our results compared to a multicenter evaluation with a higher number of individuals. In addition, our calculated sample size could not be matched mainly due to unavailable CT scans or loss to follow-up.

Moreover, our cohort consisted of patients with malignancies in different locations of the colon. Given the varying etiology and complication rates across these regions, potential biases may exist.

Additionally, the CT scan analysis was based on a single picture at the level of the umbilicus, which could lead to a generalization of the body composition of the patient. If more sections were to be examined, a more accurate depiction of the body composition could be made.

Lastly, given the multiple subgroup comparisons performed, these significant findings should be interpreted cautiously and validated in future studies.

Nonetheless, these findings demonstrate the various impacts of an altered body composition on the morbidity and outcome in patients with colon cancer. In particular, the effects on duration of surgery and length of hospital stay are important factors concerning the clinical challenges in patients with altered body composition.

Changes in nutritional intake and preoperative physical activity could modify the outcomes in affected individuals.

## 5. Conclusions

This study has shown that alterations in body composition parameters, particularly in the distribution of total and subcutaneous fat, can affect both postoperative and oncologic outcome in patients with colon cancer. Although fewer changes were observed in the overall cohort, elevated fat distribution parameters were associated with improvements in long-term survival, especially in patients with higher ASA scores. Regarding differences in muscle mass, no significant results were found.

Our findings highlight the potential for preoperative interventions to improve patients’ body constitution, thereby modifying postoperative morbidity and oncologic long-term outcome. Nutritional and muscular status should be optimized preoperatively to ensure that patients are neither underweight nor sarcopenic at the time of surgery.

## Figures and Tables

**Figure 1 jcm-14-03438-f001:**
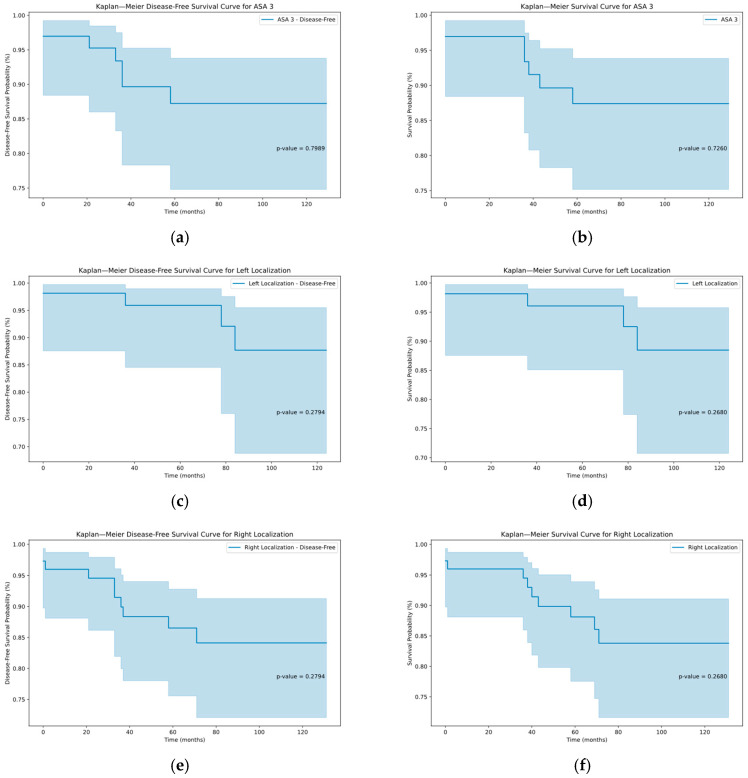
Kaplan–Meier graphs depicting the disease-free and overall survival in the subcohorts of ASA III patients (**a**,**b**), left hemicolectomy (**c**,**d**) and right hemicolectomy (**e**,**f**) with corresponding *p*-values.

**Table 1 jcm-14-03438-t001:** Body composition parameters and their gender-specific cut offs; cm centimeter; m meter; TFA total fat area; SFA subcutaneous fat area; VFA visceral fat area; SMA skeletal muscle area; SMI skeletal muscle index. Cut-offs according to [18,19].

	Male	Female
TFA (cm^2^)	>495	>392
VFA (cm^2^)	>163.8	>80.1
SFA (cm^2^)	>210	>274
VFA/TFA ratio	>0.397	>0.330
SFA/TFA ratio	>0.496	>0.715
SMA (cm^2^)	<134	<89.2
SMI (cm^2^/m^2^)	<41.6	<32

**Table 2 jcm-14-03438-t002:** Patients’ demographics; std standard deviation; ASA American Society of Anesthesiologists; UICC Union for International Cancer Control; m meter; kg kilogram; BMI body mass index; TFA total fat area; SFA subcutaneous fat area; VFA visceral fat area; SMA skeletal muscle area; skeletal muscle index; cm centimeter.

Patients’ Characteristics	Total (n = 129)
Sex, female, n (%)	52 (40.31%)
Age (year), mean ± std	71.2 ± 11.94
ASA classification, n (%)	
I	11 (8.53%)
II	48 (37.21%)
III	66 (51.16%)
IV	4 (3.1%)
V	0 (0.0%)
UICC	
0	7 (5.43%)
I	33 (25.58%)
II	51 (39.53%)
III	38 (29.46%)
IV	0 (0.0%)
pT stages, n (%)	
Tis	7 (5.43%)
T1	17 (13.18%)
T2	20 (15.5%)
T3	68 (52.71%)
T4	17 (13.18%)
pN stage, n (%)	
N0	90 (69.77%)
N1	28 (21.71%)
N2	11 (8.53%)
Comorbidities (Charlson Comorbidity Index), mean ± std	4.29 ± 1.62
Preoperative therapy, n (%)	37 (28.68%)
OP technique, n (%)	
Ileocolic resection	1 (0.78%)
Right hemicolectomy	61 (47.29%)
Extended right hemicolectomy	6 (4.65%)
Transverse resection	6 (4.65%)
Left hemicolectomy	18 (13.95%)
Extended left hemicolectomy	3 (2.33%)
Sigmoid resection	32 (24.81%)
(Sub)total colectomy	2 (1.55%)
Open	70 (54.26%)
Laparoscopic	59 (45.74%)
Conversion (yes), n (%)	11 (8.53%)
Duration of surgery (minutes), mean ± std	146.37 ± 61.06
Complications, n (%)	30 (23.26%)
Anastomotic leakage	4 (3.1%)
Wound infection	5 (3.88%)
Bleeding	4 (3.1%)
Ileus	7 (5.43%)
Renal failure	1 (0.78%)
Other complications	15 (11.63%)
Clavien–Dindo, n (%)	
0	100 (77.52%)
I	4 (3.1%)
II	8 (6.2%)
III	14 (10.85%)
IV	0 (0.0%)
V	3 (2.33%)
Length of stay (days), mean ± std	13.3 ± 6.33
Height (m), mean ± std	1.71 ± 0.1
Weight (kg), mean ± std	77.2 ± 18.54
BMI (kg/m^2^), mean ± std	26.29 ± 4.87
TFA (cm^2^), mean ± std	349.97 ± 150.1
VFA (cm^2^), mean ± std	118.2 ± 74.98
SFA (cm^2^), mean ± std	231.77 ± 100.78
VFA/TFA, mean ± std	0.32 ± 0.12
SFA/TFA, mean ± std	0.68 ± 0.12
SMA (cm^2^), mean ± std	144.84 ± 45.8
SMI (cm^2^/m^2^), mean ± std	49.37 ± 13.66
Overall survival (months), mean ± std	70.45 ± 31.19
Disease-free survival (months), mean ± std	67.48 ± 32.87
Local recurrence, n (%)	1 (0.78%)
Metastasis, n (%)	14 (10.85%)

**Table 3 jcm-14-03438-t003:** Body composition parameters and their impact, whole cohort; TFA total fat area; SFA subcutaneous fat area; VFA visceral fat area; SMA skeletal muscle area; SMI skeletal muscle index, significant *p*-values bolded.

	TFA Elevated			SFA Elevated			VFA Elevated			SFA/TFA Elevated			VFA/TFA Elevated			SMA Lowered			SMI Lowered		
	Yes	No	*p*-Value	Yes	No	*p*-Value	Yes	No	*p*-Value	Yes	No	*p*-Value	Yes	No	*p*-Value	Yes	No	*p*-Value	Yes	No	*p*-Value
Length of stay (days)	12.42 ± 4.19	13.65 ± 7.01	0.327	12.40 ± 4.80	13.82 ± 7.08	0.226	11.92 ± 4.82	14.21 ± 7.08	**0.046**	13.14 ± 5.71	13.93 ± 8.47	0.569	13.37 ± 7.52	13.27 ± 5.74	0.93	12.71 ± 5.50	13.42 ± 6.53	0.645	11.73 ± 4.56	13.51 ± 6.55	0.311
Complications	10 (27.8%)	20 (21.5%)	0.6	15 (31.9%)	15 (18.3%)	0.122	12 (23.5%)	18 (23.1%)	1	26 (25.5%)	4 (14.8%)	0.362	10 (23.3%)	20 (23.3%)	1	5 (23.8%)	25 (23.1%)	1	4 (26.7%)	26 (22.8%)	0.994
Anastomotic leakage	0 (0.0%)	4 (4.3%)	0.485	2 (4.3%)	2 (2.4%)	0.964	0 (0.0%)	4 (5.1%)	0.261	4 (3.9%)	0 (0.0%)	0.674	1 (2.3%)	3 (3.5%)	1	1 (4.8%)	3 (2.8%)	1	0 (0.0%)	4 (3.5%)	1
Overall survival (months)	72.22 ± 31.44	69.76 ± 31.40	0.691	75.11 ± 25.98	67.78 ± 33.85	0.202	75.20 ± 28.67	67.35 ± 32.73	0.165	70.63 ± 29.73	69.78 ± 37.32	0.901	72.49 ± 31.62	69.43 ± 31.29	0.603	66.33 ± 29.61	71.25 ± 31.70	0.512	60.93 ± 28.56	71.70 ± 31.56	0.212
Disease-free survival (months)	69.56 ± 32.89	66.68 ± 33.19	0.659	70.34 ± 28.71	65.84 ± 35.29	0.458	72.12 ± 30.99	64.45 ± 34.11	0.198	67.16 ± 32.01	68.70 ± 37.13	0.83	69.30 ± 33.25	66.57 ± 33.04	0.659	62.24 ± 32.25	68.50 ± 33.20	0.428	57.13 ± 32.45	68.84 ± 32.97	0.198
OP duration (min)	138.25 ± 60.84	149.62 ± 61.52	0.349	155.07 ± 64.01	141.38 ± 59.52	0.229	143.16 ± 63.98	148.56 ± 59.75	0.629	152.57 ± 63.20	123.67 ± 48.28	**0.029**	136.40 ± 59.18	151.36 ± 62.08	0.198	153.60 ± 66.76	145.01 ± 60.46	0.567	155.21 ± 75.63	145.27 ± 59.59	0.569

**Table 4 jcm-14-03438-t004:** Body composition parameters and their impact, ASA III cohort; TFA total fat area; SFA subcutaneous fat area; VFA visceral fat area; SMA skeletal muscle area; SMI skeletal muscle index, significant *p*-values bolded.

	TFA Elevated			SFA Elevated			VFA Elevated			SFA/TFA Elevated			VFA/TFA Elevated			SMA Lowered			SMI Lowered		
	Yes	No	*p*-Value	Yes	No	*p*-Value	Yes	No	*p*-Value	Yes	No	*p*-Value	Yes	No	*p*-Value	Yes	No	*p*-Value	Yes	No	*p*-Value
Length of stay (days)	11.83 ± 3.86	15.47 ± 8.44	0.055	12.54 ± 5.01	15.14 ± 8.32	0.169	11.97 ± 4.48	16.17 ± 8.79	**0.019**	14.25 ± 6.59	14.06 ± 9.31	0.925	14.33 ± 8.54	14.12 ± 6.70	0.91	12.31 ± 4.68	14.66 ± 7.84	0.305	11.33 ± 4.96	14.83 ± 7.68	0.137
Complications	7 (30.4%)/16 (69.6%)	12 (27.9%)/31 (72.1%)	1	8 (33.3%)/16 (66.7%)	11 (26.2%)/31 (73.8%)	0.738	8 (25.8%)/23 (74.2%)	11 (31.4%)/24 (68.6%)	0.817	17 (35.4%)/31 (64.6%)	2 (11.1%)/16 (88.9%)	0.102	6 (25.0%)/18 (75.0%)	13 (31.0%)/29 (69.0%)	0.817	4 (30.8%)/9 (69.2%)	15 (28.3%)/38 (71.7%)	1	3 (25.0%)/9 (75.0%)	16 (29.6%)/38 (70.4%)	1
Anastomotic leakage	0 (0.0%)/23 (100.0%)	2 (4.7%)/41 (95.3%)	0.767	1 (4.2%)/23 (95.8%)	1 (2.4%)/41 (97.6%)	1	0 (0.0%)/31 (100.0%)	2 (5.7%)/33 (94.3%)	0.527	2 (4.2%)/46 (95.8%)	0 (0.0%)/18 (100.0%)	0.942	0 (0.0%)/24 (100.0%)	2 (4.8%)/40 (95.2%)	0.734	0 (0.0%)/13 (100.0%)	2 (3.8%)/51 (96.2%)	1	0 (0.0%)/12 (100.0%)	2 (3.7%)/52 (96.3%)	1
Overall survival (months)	75.04 ± 28.88	56.70 ± 33.77	**0.031**	69.62 ± 27.91	59.36 ± 35.54	0.228	74.06 ± 30.56	53.37 ± 32.65	**0.01**	61.10 ± 31.18	68.39 ± 38.28	0.43	67.12 ± 35.16	60.79 ± 32.10	0.459	55.15 ± 32.52	65.04 ± 33.28	0.339	54.08 ± 27.89	65.09 ± 34.08	0.301
Disease-free survival (months)	73.83 ± 30.48	54.37 ± 34.54	**0.027**	66.54 ± 29.67	58.07 ± 36.58	0.338	71.19 ± 33.19	52.26 ± 33.11	**0.024**	58.44 ± 32.62	68.39 ± 38.28	0.297	64.25 ± 36.59	59.38 ± 33.15	0.582	50.77 ± 35.73	63.70 ± 33.72	0.225	49.33 ± 31.68	63.78 ± 34.51	0.188
OP duration (min)	135.09 ± 55.92	133.79 ± 49.48	0.923	159.58 ± 58.02	119.41 ± 41.04	**0.002**	140.00 ± 60.28	129.00 ± 42.01	0.393	140.45 ± 53.49	118.06 ± 42.76	0.117	130.67 ± 57.68	136.34 ± 48.02	0.671	133.92 ± 59.37	134.33 ± 49.88	0.98	135.92 ± 60.67	133.87 ± 49.75	0.902

**Table 5 jcm-14-03438-t005:** Body composition parameters and their corresponding hazard and odds ratio, whole cohort; TFA total fat area; SFA subcutaneous fat area; VFA visceral fat area; SMA skeletal muscle area; SMI skeletal muscle index; HR hazard ratio; OR odds ratio; na not applicable, significant *p*-values bolded.

	TFA Elevated			SFA Elevated			VFA Elevated			SFA/TFA Elevated			VFA/TFA Elevated			SMA Lowered			SMI Lowered		
	HR	OR	*p*-Value	HR	OR	*p*-Value	HR	OR	*p*-Value	HR	OR	*p*-Value	HR	OR	*p*-Value	HR	OR	*p*-Value	HR	OR	*p*-Value
Length of stay (days)	na	1.36 (0.63–2.94)	0.055	na	0.85 (0.41–1.74)	0.169	na	0.98 (0.49–2.00)	**0.019**	na	0.54 (0.23–1.28)	0.925	na	1.10 (0.53–2.28)	0.91	na	1.02 (0.40–2.59)	0.305	na	0.97 (0.33–2.86)	0.137
Complications	na	1.42 (0.60–3.38)	1	na	2.08 (0.92–4.71)	0.738	na	1.03 (0.45–2.35)	0.817	na	1.81 (0.60–5.44)	0.102	na	1.02 (0.43–2.38)	0.817	na	1.09 (0.38–3.16)	1	na	1.31 (0.40–4.22)	1
Anastomotic leakage	na	0.27 (0.01–5.19)	0.767	na	1.77 (0.30–10.60)	1	na	0.16 (0.01–3.05)	0.527	na	2.51 (0.13–48.11)	0.942	na	0.84 (0.12–5.91)	0.734	na	2.21 (0.31–15.82)	1	na	0.79 (0.04–15.44)	1
Overall survival (months)	1.89 (0.66–5.46)	0.85 (0.39–1.85)	**0.031**	0.90 (0.30–2.69)	0.97 (0.47–1.99)	0.228	0.56 (0.17–1.78)	1.12 (0.55–2.27)	**0.01**	**1.59 (0.35–7.12)**	0.66 (0.28–1.54)	0.43	1.04 (0.35–3.11)	1.45 (0.70–3.03)	0.459	1.40 (0.39–5.01)	1.28 (0.50–3.25)	0.339	1.37 (0.31–6.14)	0.72 (0.24–2.14)	0.301
Disease-free survival (months)	1.35 (0.45–4.04)	0.88 (0.41–1.89)	**0.027**	2.79 (0.93–8.32)	0.73 (0.36–1.51)	0.338	1.03 (0.36–2.98)	1.25 (0.61–2.53)	**0.024**	**3.31 (0.43–25.33)**	0.61 (0.26–1.44)	0.297	1.10 (0.37–3.28)	1.67 (0.80–3.51)	0.582	2.21 (0.69–7.06)	1.14 (0.45–2.91)	0.225	2.40 (0.67–8.61)	0.64 (0.22–1.93)	0.188
OP duration (min)	na	0.58 (0.26–1.28)	0.923	na	1.46 (0.70–3.01)	**0.002**	na	0.61 (0.30–1.26)	0.393	na	2.21 (0.91–5.40)	0.117	na	0.62 (0.29–1.31)	0.671	na	1.08 (0.41–2.80)	0.98	na	0.78 (0.25–2.39)	0.902

**Table 6 jcm-14-03438-t006:** Body composition parameters and their corresponding hazard and odds ratio, ASA III cohort; TFA total fat area; SFA subcutaneous fat area; VFA visceral fat area; SMA skeletal muscle area; skeletal muscle index; HR hazard ratio; OR odds ratio; na not applicable.

	TFA Elevated			SFA Elevated			VFA Elevated			SFA/TFA Elevated			VFA/TFA Elevated			SMA Lowered			SMI Lowered		
	HR	OR	*p*-Value	HR	OR	*p*-Value	HR	OR	*p*-Value	HR	OR	*p*-Value	HR	OR	*p*-Value	HR	OR	*p*-Value	HR	OR	*p*-Value
Length of stay (days)	na	0.61 (0.22–1.75)	0.055	na	0.73 (0.26–2.03)	0.169	na	0.58 (0.22–1.57)	0.019	na	1.22 (0.40–3.69)	0.925	na	0.73 (0.26–2.03)	0.91	na	1.21 (0.36–4.09)	0.305	na	0.96 (0.27–3.42)	0.137
Complications	na	1.15 (0.39–3.39)	1	na	1.41 (0.48–4.11)	0.738	na	0.77 (0.27–2.21)	0.817	na	3.67 (0.86–15.68)	0.102	na	0.77 (0.26–2.31)	0.817	na	1.18 (0.33–4.18)	1	na	0.86 (0.22–3.33)	1
Anastomotic leakage	na	0.35 (0.02–7.67)	0.767	na	1.77 (0.17–17.95)	1	na	0.21 (0.01–4.61)	0.527	na	1.99 (0.09–43.45)	0.942	na	0.33 (0.02–7.18)	0.734	na	0.76 (0.03–16.85)	1	na	0.84 (0.04–18.62)	1
Overall survival (months)	0.25 (0.03–2.04)	2.60 (0.91–7.44)	0.031	0.56 (0.11–2.91)	1.00 (0.37–2.73)	0.228	0.00 (0.00–inf)	3.08 (1.13–8.41)	0.01	2.13 (0.26–17.72)	0.39 (0.13–1.21)	0.43	0.29 (0.03–2.40)	2.22 (0.80–6.21)	0.459	0.74 (0.09–6.17)	0.83 (0.24–2.79)	0.339	0.80 (0.10–6.66)	0.66 (0.19–2.35)	0.301
Disease-free survival (months)	0.58 (0.11–2.99)	3.50 (1.19–10.28)	0.027	3.54 (0.69–18.29)	1.00 (0.37–2.73)	0.338	0.68 (0.15–3.06)	3.08 (1.13–8.41)	0.024	13450535.01 (0.00–inf)	0.39 (0.13–1.21)	0.297	0.71 (0.14–3.66)	1.69 (0.61–4.67)	0.582	3.87 (0.86–17.31)	0.83 (0.24–2.79)	0.225	4.19 (0.93–18.82)	0.66 (0.19–2.35)	0.188
OP duration (min)	na	0.77 (0.28–2.14)	0.923	na	3.47 (1.20–10.01)	0.002	na	0.64 (0.24–1.71)	0.393	na	2.27 (0.73–7.07)	0.117	na	0.52 (0.19–1.45)	0.671	na	0.93 (0.27–3.13)	0.98	na	0.78 (0.25–2.39)	0.902

## Data Availability

The original contributions presented in this study are included in the article. The datasets generated and/or analyzed during the current study are available from the corresponding author upon reasonable request.

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
