# Peer review of "Alterations in Body Composition Lead to Changes in Postoperative Outcome and Oncologic Survival in Patients with Non-Metastatic Colon Cancer"

_jcm, 2025, doi:10.3390/jcm14103438_

Round 1
Reviewer 1 Report
Comments and Suggestions for Authors
Reviewer comments
- Add the sample size calculation in the methods section before statistical methods.
- Text formatting has several typographical (e.g, line numbering, misplaced symbols). Please correct this before final submission.
Figures: Kaplan-Meier curves (Figure 1) are referenced but not fully visible in the provided version please ensure high-resolution and put the p value inside the figure .
Conduct multivariate Cox regression analysis for survival endpoints to adjust for: Age, gender, comorbidity, cancer stage, surgical approach (laparoscopic vs open) , preoperative therapy
- Provide valid references for the cutoffs for SFA, VFA, SMA, and derived ratios.
- Report effect sizes (HRs, ORs) with 95% confidence intervals, even if p-values are non-significant.
Author Response
Dear Reviewer,
Thank you very much for your review. In the following you will find our responses:
Comment 1: Add the sample size calculation in the methods section before statistical methods.
Response 1: Sample size calculation was added in the recommended section
Comment 2: Text formatting has several typographical (e.g, line numbering, misplaced symbols). Please correct this before final submission.
Response 2: The format has been revised and corrected
Comment 3: Figures: Kaplan-Meier curves (Figure 1) are referenced but not fully visible in the provided version please ensure high-resolution and put the p value inside the figure .
Response 3: The Kaplan Meier curves are updated with a higher resolution and the p-values are added
Comment 4: Conduct multivariate Cox regression analysis for survival endpoints to adjust for: Age, gender, comorbidity, cancer stage, surgical approach (laparoscopic vs open) , preoperative therapy
Response 4: We did conduct a multivariate regression analysis, but it did not conclude in significant results. Further description in methods and results sections were added now.
Comment 5: Provide valid references for the cutoffs for SFA, VFA, SMA, and derived ratios.
Response 5: References for the cut offs added
Comment 6: Report effect sizes (HRs, ORs) with 95% confidence intervals, even if p-values are non-significant
Response 6: Tables 5 & 6 with OR & HR added
Thank you once more for your comments.
Best regards,
Markus Weigl, MD
Reviewer 2 Report
Comments and Suggestions for Authors
This manuscript addresses an important and timely topic: the role of body composition in surgical outcomes and cancer survival. The content is rich with data and the authors have clearly put effort into analyzing both the whole cohort and subgroups. The key critique points revolve around ensuring clarity in how results are presented (especially when they defy expectations), justification of methodological choices, and careful interpretation of the surprising findings. The authors should be commended for investigating multiple facets of body composition (fat distribution and muscle) rather than just BMI. With the revisions suggested – such as clarifying cut-off derivations, adding explanations for paradoxical trends, and tightening the conclusion – the manuscript will be scientifically robust and easier for readers to follow. The language and formatting issues are minor and not the focus here; content-wise, the study is valuable, and addressing these comments will help sharpen its impact and credibility.
Introduction
- authors state the aim of the study (to investigate the influence of fat and muscle parameters on outcomes in non-metastatic colon cancer), which is good. However, they could better frame a specific hypothesis or clarify the gap in knowledge. For example, it would help to mention whether prior studies in colon cancer patients have shown inconsistent or unclear results regarding body composition (especially since numerous studies in colorectal cancer overall have linked sarcopenia and visceral obesity to worse outcomes)
- it might be worth briefly noting that colon cancer patients typically do not receive preoperative chemoradiation , which could mean body composition influences might differ between colon and rectal cancers; a one-sentence hint of this context would underline the novelty of focusing on colon cancer specifically
Materials and Methods
- Patients without a CT were excluded. The authors should acknowledge how many patients were omitted due to lack of CT and consider whether these excluded patients had different characteristics. Since this is retrospective, it’s possible that only certain patients got pre-op CTs. This could skew the cohort. Mentioning this limitation and perhaps comparing included vs excluded baseline data would strengthen the validity of the method.
- The authors state they took a single axial slice at the level of the umbilicus to measure fat and muscle areas. Typically, studies use the L3 vertebral level as a standard for body composition assessment; using the umbilicus level is acceptable if validated, but the authors should clarify why this level was chosen. Was it because it was easier to standardize or retrieve?
- Additionally, how were visceral fat area (VFA) and subcutaneous fat area (SFA) distinguished on that slice?
- The manuscript mentions HU thresholds, but it doesn’t explicitly describe separating visceral from subcutaneous compartments. Readers might assume that the software or an analyst segmented the abdominal wall boundary to differentiate visceral vs subcutaneous fat. It would be helpful to state that manual or semi-automated segmentation was done to isolate intra-abdominal (visceral) fat from subcutaneous fat, so that VFA and SFA were measured separately on the same slice
- The justification for cut-off thresholds should be explicitly stated. Did the authors derive them from a reference population or prior studies (if so, cite those sources), or are they based on the sample’s own distribution ( median values or upper quartile)?
- The methods mention that a combination of low muscle and high fat (likely low SMI with high VFA) was considered “sarcopenic obesity.” It’s good that the authors attempted to examine this combined phenotype, as it is known to confer risk in some studies. However, ensure this definition is clear: “Patients with SMI below the cut-off and VFA above the cut-off were categorized as having sarcopenic obesity.” Also, clarify if any specific analysis was done on this subgroup (it’s hinted in Discussion comparing to Fleming et al.’s findings). If sarcopenic obesity was analyzed, describe how (did you compare those with vs without it on outcomes?). This might be missing from the current Methods description.
- Histopathological Examination: The methods briefly note that resected specimens were examined and staged (TNM classification [18]). This is straightforward. One minor addition could be to clarify how follow-up data on oncologic outcomes were obtained. For example, was follow-up done through clinic visits or a cancer registry to determine disease-free and overall survival times? Since survival is an endpoint, a sentence on follow-up procedure (and median follow-up time) would be useful. If this information is presented later in Results, it could be referenced here.
- Statistical Analysis: It’s commendable that the analysis was done using a modern platform (Python 3.9.1) and a variety of statistical tests. The authors appropriately used non-parametric tests (Mann-Whitney U, Kruskal-Wallis) presumably when normality was in doubt, and t-tests otherwise; chi-square for categorical comparisons; and Kaplan-Meier survival analysis with log-rank tests for survival outcomes. A few concerns or suggestions here:
- Were any multivariate analyses performed? If no multivariate analysis was done, consider stating that explicitly and perhaps justifying it (maybe the sample size of 129 is borderline for including many covariates). As is, readers might wonder if the observed survival differences could be explained by, say, the fact that patients with low fat might have had higher stage disease or were older
- The subgroup analysis by ASA class III should be introduced in the Methods if it was a pre-planned analysis. In the Results, the authors devote a lot of attention to patients with ASA III status. If this subgroup was of interest from the start (which makes sense, since ASA III was the largest group and perhaps more homogeneous in health status), the Methods should mention something like “We performed a subgroup analysis focusing on the subset of patients with ASA classification III, to see if body composition effects were more pronounced in a higher-risk population.” This would prepare the reader for the detailed ASA III results later. If it was not pre-specified but done post-hoc, the authors should be transparent about that too. In either case, clarifying the plan for subgroup analyses helps ensure the reader that the analysis wasn’t an afterthought or data dredging.
- Minor point: The authors might mention how survival time was calculated (from date of surgery to event or last follow-up) and how they handled censored data (standard Kaplan-Meier methodology, presumably). It’s mostly obvious, but a sentence for completeness would be fine.
Results
- Tumor stages are given (nearly 40% stage II and ~55% stage III, which implies only ~5% were stage I). The text says “nearly half were UICC stage II (39.53%)” – that’s fine, though one could also note a slight majority were stage III. It’s a minor wording preference, but not a big issue.
- It’s reported that 54.26% underwent open surgery (n=70) and 45.74% laparoscopic (n=59), with 11 patients needing conversion from laparoscopy to open. There is a slight ambiguity here in how these numbers are reported. By counting final approach, it appears 70 ended up as open surgeries (including those converted) and 59 were completed laparoscopically. It might help to clarify: “70 patients (54%) ultimately had open procedures (including 11 cases converted from laparoscopy), while 59 (46%) were completed laparoscopically without conversion.”
- The mean overall survival (OS) was 70.45 ± 31.19 months, and mean disease-free survival (DFS) was 67.48 ± 32.87 months. This suggests that, on average, patients were followed for roughly 5–6 years after surgery. It would help to report the median follow-up time or median survival if available, because means with such wide standard deviations are hard to interpret (some patients likely had much shorter follow-up, others very long).
- I’d recommend reporting 5-year OS and DFS rates or median survival if appropriate, as those are more standard oncologic metrics. This is a suggestion to improve how the results are communicated, though not strictly necessary if space is an issue
- The authors report no significant differences in overall complication rate or in AL incidence between patients with “normal” vs “altered” body composition parameters. In other words, whether a patient had high vs low fat or muscle did not significantly affect whether they had postoperative complications in this cohort. This negative finding is important: it contrasts with some prior studies that have found obesity or sarcopenia to be risk factors for complications. It would be helpful if the authors provided a bit more data here for the reader’s benefit, even if not all differences reached significance. For example, they could mention “the overall complication rate was similar between sarcopenic vs non-sarcopenic patients (p=…)” or whichever comparisons they did check.
- It is reported that patients with an elevated total fat area (TFA) had a shorter hospital length of stay on average (11.92 ± 4.82 days) compared to those with lower TFA (14.21 ± 7.08 days), and this was statistically significant (p = 0.046). This is an interesting finding, suggesting that higher overall adiposity might have been associated with faster recovery or fewer reasons to keep the patient hospitalized. One could interpret that perhaps very low-fat patients had more protracted recoveries. The result is borderline significant (p just under 0.05), so it should be interpreted cautiously. Still, it’s a noteworthy piece of data that goes against a simplistic expectation that “more fat = slower recovery.” The authors should ensure this result is accurately represented in Table 3 and perhaps verify that no other confounding factor (like maybe those with high TFA were younger?) is driving it. But as a result, it’s clearly stated and will be good to discuss as part of the so-called “obesity paradox.”
- Operative Duration and Fat Distribution - One suggestion: clarify whether this was observed in all surgeries or specifically in open vs laparoscopic. If most obese patients ended up getting open surgeris, that could influence operative time independently. The data as presented treat it as a simple two-group comparison by SFA/TFA ratio, which is fine. Just be careful in phrasing – an elevated ratio “was associated with” longer surgery
- It’s worth noting (as a critique) that the study tested quite a few parameters (TFA, VFA, SFA, two fat ratios, SMA, SMI) against multiple outcomes (complications, AL, LOS, op time, OS, DFS, in two cohorts – overall and ASA III). This amounts to a large number of comparisons. The manuscript does not mention any correction for multiple hypothesis testing, so there is a risk of false-positive findings. The fact that some p-values (like 0.024, 0.027, 0.041) are modestly significant means they could potentially arise by chance. On the other hand, the consistency of the direction (all pointing to “more fat = better long-term outcome in ASA III”) gives some credence that it’s real. The authors might want to acknowledge this issue. They don’t necessarily need to adjust p-values (which would likely render most insignificant given the small sample), but a statement like “we note that we performed multiple subgroup comparisons, so these findings, while significant, should be interpreted with caution and need confirmation” would be prudent.
- As mentioned, double-checking the numbers between tables and text is important. For example, the confusion between using “median” vs “mean” for age should be fixed for accuracy. Also, the duplicate value for OS and DFS low-TFA group in text is likely a typo. Such small errors can distract readers and cast doubt on the rigor, even though they are easy to correct. Since this review focuses on content, the main point is to ensure accuracy in reporting the data.
Discussion
- The authors mention another study [21] where loss of muscle during neoadjuvant therapy in rectal cancer led to worse oncologic outcomes – again highlighting that in rectal cancer, sarcopenia can be a big issue, possibly due to treatment effects. They correctly note that since colon cancer typically doesn’t use radiochemotherapy preoperatively, this might partly explain why their colon cohort didn’t show the same associations. This is a plausible explanation and it’s good to see them think about biological differences between the two diseases. They might consider explicitly using the term “obesity paradox” when describing how, in their colon cancer patients (especially ASA III), higher fat correlated with better outcomes.
- The authors note that prior studies have been inconsistent about whether obesity prolongs hospital stay. They cite one study [24] that showed a decrease in hospital stay with higher BMI, and other studies showing no impact. One suggestion: add that the difference in LOS was on the order of ~2-3 days and occurred in ASA III patients, who likely have lower physiological reserve, amplifying the impact of any extra surgical stress. That could tie together why it was seen in ASA III but not in the overall group (where maybe healthier ASA I-II patients compensated better even if visceral fat made surgery harder).
Conclusion
- Saying “alterations in body composition parameters… affect both postoperative and oncologic outcome” is a strong statement. Given that in the overall analysis most postoperative outcomes were not affected, and oncologic outcomes were only affected in a subset, the authors might soften this to “may affect” or “can affect”.
- The conclusion currently focuses on fat distribution and fat-related findings. It does not mention that skeletal muscle index (SMI) was not significantly associated with outcomes. Since the study set out to examine muscle mass as well, it would be honest to include a note about that. Perhaps something like: “Notably, in this cohort low skeletal muscle mass alone was not significantly linked to worse outcomes, highlighting that fat-related metrics might be more relevant in this population.”
- Just ensure any logical gaps are closed: for example, if a reader wonders “so should we fatten up patients to improve survival?” – the conclusion should implicitly answer “not exactly, but we should optimize nutrition/muscle such that patients are not underweight or sarcopenic going into surgery.” Perhaps adding a clause to clarify that what’s beneficial is likely avoiding poor nutritional status. The authors allude to this by mentioning interventions and body “constitution” improvements.
Good job! Cheers!!!
Author Response
Dear Reviewer,
thank you for your comments to improve this manuscript. We will try my best to answer the remarks you made:
Comment 1: Introduction
Response 1:
- like you mentioned, the knowledge gap, that we aimed to close was described
- Comment regarding non neoadjuvant RCT in colon cancer patients added
Comment 2: Methods
Response 2:
- all patients that are operated in our clinic do get pre op CT scans. Only a small number of individuals had to be excluded (9), primarily because in this cases the CTs were conducted in an external clinic/practice and could not be restored in our system.
- Concerning the level of the CT scan, we did a similar study with patients with rectal cancer (DOI 10.3390/nu15112632) last year and during the research process we evaluated the most common level of CT scans that were used in evaluating BCP. Like you mentioned L3 is also used regularly but in our research the umbilicus was used as often if not more often.
- During the CT scan evaluation first the TFA was evaluated, afterwards the intra abdominal content was removed and the SFA was evaluated. Lastly SFA was subtracted from TFA thus resulting in the VFA
- Sources for the BCP cut offs were added
- comments regarding sarcopenia obesity were acknowledged and added
- the method of follow up conduction was added in methods
- a multivariate analysis was conducted but resulted in no significant results. Descriptions in methods and results were added.
- Descriptions regrading the subgroup analysis were included
- DFS and OS were disrupted
Comment 3: Results:
Response 3:
- wording concerning UICC stage and conversions was adapted
- Median OS and DFS were added
- additional description concerning TFA/SMA and complication rates added
- regarding the shorter LoS in patients with higher VFA we found no secondary factor (like you mentioned e.g. younger age)
- clarifying wording regarding the longer surgery time in patients with elevated SFA was added
- a limitation sentence was added in the discussion regarding the amount of our analysis
- change from median to mean in the age description conducted
- regarding the possible double value, it was one time the higher TFA resulted in a better survival, the second one is regarding VFA and its improvement on survival
Comment 4: Discussion
Response 4:
- thank you for the remark regarding the differences in colon and rectal cancer and the role of RCT in the different outcomes of these two
- further clarification concerning the results in ASA III patients added
Comment 5: Conclusion
Response 5:
- introduction sentence changed
- remarks concerning the no significant results in SMI/SMA patients were added
- an explanatory last sentence concerning a recommendation was added
Thank you once more for your comments. We hope we answered them as you intended.
Best regards
Markus Weigl, MD